# The Experience of VaccinarSinToscana Website and the Role of New Media in Promoting Vaccination

**DOI:** 10.3390/vaccines8040644

**Published:** 2020-11-03

**Authors:** Sara Boccalini, Paolo Bonanni, Fabrizio Chiesi, Giulia Di Pisa, Federica Furlan, Barbara Giammarco, Beatrice Zanella, Francesco Mandò Tacconi, Angela Bechini

**Affiliations:** 1Department of Health Sciences, University of Florence, 50134 Florence, Italy; sara.boccalini@unifi.it (S.B.); paolo.bonanni@unifi.it (P.B.); beatrice.zanella@unifi.it (B.Z.); 2Medical Specialization School of Hygiene and Preventive Medicine, University of Florence, 50134 Florence, Italy; fabrizio.chiesi@unifi.it (F.C.); giulia.dipisa@unifi.it (G.D.P.); federica.furlan@unifi.it (F.F.); barbara.giammarco@unifi.it (B.G.); 3Nuovo Ospedale delle Apuane, North-West Tuscany LHU (Local Health Authorities), Viale Enrico Mattei, 21, 54100 Marina di Massa, Italy; francesco.mandotacconi@uslnordovest.toscana.it

**Keywords:** vaccination, information, website, VaccinarSi, VaccinarSinToscana, new media, Italy

## Abstract

The Department of Health Sciences (University of Florence) developed a regional website “VaccinarSinToscana” in order to provide information on vaccines and communicate with the general population, as well as the healthcare community, at a regional and local level. The aim of this paper is to present the VaccinarSinToscana website framework and analyze the three-year activity of the website and the related social network account on Facebook in terms of dissemination and visibility. In the first three years since its launch, the VaccinarSinToscana portal has increased its visibility: the number of single users, visits and single web pages has grown exponentially. Our results also demonstrate how the Facebook account launch contributed enormously to the increase in the visibility of the website. The objective for the future of the VaccinarSinToscana portal is to grow further, in order to reach out to an even wider audience.

## 1. Introduction

Vaccines are among the greatest public health achievements of the last two centuries. They are estimated to save 2–3 million lives each year [1]. Vaccines have successfully eradicated smallpox and have greatly reduced the incidence of several major diseases such as poliomyelitis and measles. [2] Licensed vaccines are now available to prevent over 30 different infectious diseases, several of which can be combined or co-administered during the same vaccination visit [3]. Targets of prevention are not only children: indeed, elderly people, adolescents, pregnant women, people suffering from chronic and immune-compromising diseases represent new target groups. Vaccines can even help to prevent cancer [1,4]. To date, the success of vaccinations has made possible the reduction in burden due to preventable infectious diseases.

As people have no longer had contact with these diseases, the fear of adverse reactions to vaccines has increased and a new resumption of “vaccine hesitancy” movements has been seen in recent years. Parents and patients, worried more for vaccine risks than for the disease and related complications, are choosing to delay or refuse vaccines or asking for alternative vaccination schedules [5,6,7,8]. Having no experience with preventable diseases, parents do not feel the need to immunize their children. At the same time, trust in institutional medicine has decreased. Actually, the strategy to talk to parents with an open-ended or presumptive approach, or with motivational interviewing to reduce the knowledge gap in patients with educational materials or other methods, do not seem to be effective at all. Relying on the provider–patient relationship and on the policy of vaccination as a school entry mandate seems to give results [9]. On the other hand, anti-vaccination groups have improved their communication techniques: blogs, internet websites, media, social-media, YouTube videos, famous testimonials [9].

In this scenario, to limit vaccine hesitancy is fundamental to involve primary care physicians and pediatricians in these new tools. In order to identify, plan and implement effective communication strategies, healthcare professionals should be trained and skilled in innovative methods and new media [10]. Particularly, web-based interventions and social media tools could be now considered promising approaches [11]. As an example, the European Vaccine Today project (which has developed an online platform to inform and discuss on vaccination in March 2011) established that tools such as Google AdWords or Facebook advertising are useful to address web users, searching for vaccine information, towards websites that spread valuable scientific content [12]. In addition, the ESCULAPIO project, funded by the Italian Ministry of Health, identified the important role of these new information sources, in order to address adequate strategies to improve information on vaccines and vaccine preventable diseases [13].

In Italy, the Italian Society of Hygiene and Preventive Medicine (SItI) developed in 2013 the national VaccinarSì Project [14]. This project received the institutional patronage of the Italian Ministry of Health and initiated collaborations with the other Italian scientific societies involved in immunization programs and policies, as the Italian Society of General Practice and Primary Care (SIMG), the Italian Federation of Family Pediatricians (FIMP) and the Italian Federation of General Practitioners (FIMMG). Following the success of the National portal [14,15], some regions began to develop a regional version of the website. The current 10 regional websites (VaccinarSinVeneto, VaccinarSinToscana, VaccinarSinPuglia, VaccinarSinSicilia, VaccinarSinLiguria, VaccinarSinSardegna, VaccinarSinCampania, VaccinarSinLazio, VaccinarSinelleMarche and VaccinarSinTrentino) [16] provide reports with clear, coherent and scientifically reliable information on issues of territorial relevance and respond to local information needs. In fact, in Italy, it is strategic to have regional sites that provide local information, because the regions have some degree of autonomy in the management of health systems, particularly in the prevention activities (such as the writing of regional vaccination calendar and organization of immunization centers).

One of the regional websites (VaccinarSinToscana) was developed by the Department of Health Sciences (University of Florence). The goal of this website is to communicate and educate the general population, as well as the healthcare community, about vaccines using the new social media. A specific aim of the website is to produce and disseminate evidence-based, solid, comprehensive, understandable, and updated information about vaccines, counterbalancing the misleading and erroneous information circulating on the web, in order to increase vaccination confidence. In particular, among the principal activities of the website is to provide information on vaccines and communicate to the population at a regional and local level.

The aim of this paper is to present the VaccinarSinToscana website framework; show all the activities carried out in order to ensure correct information, especially at the local level; describe the socio-demographic characteristics of the users (gender, age); and analyze the three-year activity of the website and the related social network account on Facebook with regard to dissemination and visibility, which were assessed in terms of number of single users, visits and single web pages visited. Therefore, we expect that more and more users will access the website VaccinarSinToscana in the future and that the website will become an example of an active tool to spread correct information about vaccinations in the general population.

## 2. Materials and Methods

The framework of VaccinarSinToscana is presented as organizational staff, institutional patronage and funding. The website structure and contents are also described. The website activity in the study period 21st March 2017–20th March 2020 is obtained from Google Analytics. Google Analytics allows the analysis of the number and the socio-demographic characteristics (gender, age and geographical location) of single users who accessed the website, the number of visits and single web pages visited. The social network account activities are collected from Facebook Insights. Facebook Insights allows the analysis of the number and the socio-demographic characteristics (gender, age and geographical location) of followers. A further analysis was performed to present activities during the period of the COVID-19 emergency phase in Italy (21st March 2020–20th June 2020).

## 3. Results

### 3.1. Organizational Staff

The VaccinarSinToscana website organizational structure includes a scientific committee, an operational board and a communication task force. The scientific committee includes 24 experts in the field of immunization from the academia and regional health services that collaborate on a voluntary basis and without conflicts of interest. The coordinator is responsible for defining the guiding strategies of the website and validating the portal’s scientific content. The operational board comprises doctors attending the Specialization School in Hygiene and Preventive Medicine of the University of Florence and is in charge to maintain, update and monitor the portal. The communication task force comprises four residents in Hygiene and Preventive Medicine of the University of Florence and is in charge of monitoring the portal’s content, publishing the news, updating the social network account (Facebook) and answering emails from the audience.

### 3.2. Institutional Patronage and Funding

The VaccinarSinToscana website was initially funded by the Italian Ministry of Health, through a competitive call of the Italian Center of Disease Control in 2014 (CCM-2014) and actually is funded by the Department of Health Sciences (University of Florence). The website received the institutional patronage of the Tuscany Region. Members of the scientific committee, the operational board and the communication task force work at the VaccinarSinToscana website on a voluntary basis.

### 3.3. VaccinarSinToscana Website Structure and Contents

The website is open and freely accessible to the public. The portal includes four different sections, the same as the national and the other regional websites. The first section reports National and Regional Policies about vaccination and the updated Regional Vaccination Calendar. The second describes vaccine-preventable diseases, available vaccines, vaccine benefits and risks, scientific evidences against misinformation on vaccinations and recommendations for international travelers. The third provides up-to-date information on vaccination clinics in the regional territory: particularly, information on addresses, opening hours and contacts. The fourth is the section dedicated to News and Events. The News section includes brief texts, reporting on recent published scientific studies, guidelines, immunization schedules, recently released epidemiological data and infectious disease outbreaks of international, national and regional relevance. The Event section lists relevant regional conferences, congresses and seminars in the field of immunization and infectious diseases.

The VaccinarSinToscana social network account on Facebook reports the website updates with a direct connection with the portal.

### 3.4. VaccinarSinToscana Website and Facebook Account Activities

The VaccinarSinToscana website was launched on 21st March 2017. In the three-year study period (21st March 2017–20th March 2020), about 52,000 new single users accessed the portal, and nearly 15% of them accessed it more than once, for more than 65,000 visits. Overall, around 104,000 single web pages were accessed, with 1.58 web pages visited on average for each access to the portal and an average session duration of one minute and two seconds.

With regard to socio-demographic characteristics, 67% of users were female and the 25–34 year age group accounted for 31%. Other age groups accounted for 28% (35–44 years), 17% (45–54 years), 9% (55–64 years), 8% (18–24 years) and 7% (over 64 years). The vast majority of users were from Italy (97%). Access from other 110 nations were reported; the most represented nations were USA, UK and Germany. Within Italy, Florence (26%), Milan (12%) and Rome (9%) were the cities with the higher number of reported access to the portal. In Tuscany, besides Florence, Prato (6%), Lucca (5%) and Pisa (3%) were the highest-ranking cities. Mobile phones were the most widely used devices to access the portal (66%). Personal computers and tablets accounted for, respectively, 31% and 3%.

The majority of visitors (85%) access the VaccinarSinToscana portal through organic search (mainly Google), 7% access directly the website, 5% through a referral website (mainly Tuscany Region portal) and 3% through social network (mainly Facebook). With regard to specific webpage visits, the most visited pages were those on the Regional Vaccination Calendar and vaccination clinics in Florence with about 7300 visits each. The Papilloma Virus vaccination data sheet was the most widely read with about 2700 visits and the article about the 10 most common vaccine fake news was the most widely read with about 2500 visits.

The number of annual single users, visits and single web pages registered by the VaccinarSinToscana website increased during the three years of the study (Figure 1). The increase is particularly evident in the last year.

During the first year, 2061 new single users accessed the portal, representing 4% on the total of three-year single users. This number increased to 11,866 during the second year (23%), until it reached 38,096 during the third year (73%). The number of website visits went from 2456 in the first year (4%) to 14,602 in the second year (22%) and up to 48,798 in the third year (74%). The number of single web pages also reflected the same increase, going from 5246 in the first year (5%) to 25,713 in the second year (25%) and to 73,064 in the third year (70%).

The number of website visits per user was 1.19 during the first year, 1.23 during the second year and 1.27 during the third year. The number of single web pages per visit was 2.14 during the first year, 1.76 during the second year and 1.50 during the third year. Finally, the number of single web pages per user was 2.55 during the first year, 2.17 during the second year and 1.92 during the third year.

In the last year, the portal recorded an increase in e-mail correspondence. In particular, during the last analyzed semester, the communication task force answered about 100 e-mails from 70 different users. The most frequently request concerned the timetables and contacts of territorial vaccination clinics and how parents can obtain the vaccination certificate for their children.

We analyzed the monthly trend of number of single users, visits and single web pages during the last year, from April 2019 to February 2020. (Figure 2).

In the summer period (June–August 2019), the trend of the number of single users, visits and single web pages decreased, while it increased starting from September 2019 reaching a peak in October–November 2019 and January 2020.

During the COVID-19 emergency phase (21st March 2020–20th June 2020), the portal recorded 13,557 single users, 16,384 website visits and 21,795 single web pages. The number of website visits per user was 1.21, the number of single web pages per visit was 1.33 and the number of single web pages per user was 1.61.

The VaccinarSinToscana Facebook account was launched in October 2018. The account reached 487 “likes” and 497 followers in the study period. The majority of followers are females (60%) and 25–34 years old (39%). Other age groups accounted for 26% (35–44 years), 17% (45–54 years), 9% (55–64 years), 5% (over 64 years) and 4% (18–24 years). Females are also reported to be more active sharing information and commenting posts. Almost all followers (97%) were Italian. The remaining followers came from 10 different nations, among which the most represented were France, USA and the Netherlands. In Tuscany, 27% of followers came from Florence and 18% from Prato.

The number of annual single users, visits and single web pages registered by the VaccinarSinToscana website has greatly increased after the launch of the Facebook account (Figure 3).

Before the Facebook account was launched, 4997 new single users accessed the VaccinarSinToscana for a total of 6517 visits and 13,097 single web pages. After the Facebook account was launched, 46,969 new single users accessed for a total of 59,699 visits and 90,926 single web pages. The increase was about 9–10 times as regards the number of new single users and visits and about 7 times as regards the number of single web pages.

## 4. Discussion

In 2019, about 74.7% of families in Italy have been using the internet from home connection and young people connect with smartphones every day [17]. People use the internet frequently in order to search for all kinds of information including health topics and vaccines. The quality of information is variable with a substantial amount of inaccurate or misleading messages, which seem to increase negative attitude more than websites pro-vaccination do to the increase in positive beliefs. As a matter of fact, the anti-vaccination groups use this approach to spread their theories. Nowadays, anti-vaccine groups are mostly well-educated, composed of middle- and upper-income parents who want to make an “informed decision” and to label themselves as neutral instead of “no-vax” people [7].

There are many factors contributing to hesitancy: wrong perception of vaccine-preventable disease risk (based on previous experiences or on a lack of experience), access to information and misinformation, exposition at media and social-media, social norm, trust (or lack of) in health workers and public health policy. In this context, web-based content is not always well regulated, and the spread of erroneous and misleading information cannot be limited or monitored at all [18]. Thus, even if nowadays there are many approaches (i.e., storytelling and messages delivered by testimonials) [19] to encourage people to accept vaccination, among the new available tools is the use of institutional websites promoted by scientific societies.

In the first three years since its launch, the VaccinarSinToscana portal has increased its visibility: indeed, the number of single users who accessed the website, the number of visits and single web pages visited have grown exponentially. This result could be related to the growing need to find health information on the internet, especially about vaccination. Particularly, these data were confirmed by the increasing number of visits to the regional website during the COVID-19 emergency phase, when in a three-month period, the number of single users reached almost half of the total number of those registered in the whole previous year.

Therefore, the VaccinarSinToscana website represents an institutional way to promote the prevention culture. The strength of the portal is to publish evidence-based topics, with a special focus on local and regional information. The sources of information are always verified and reported by experienced personnel. Moreover, facts are shown in a clear and referenced way to be easily understood and retrieved by the general population.

The peaks in the number of single users, visits and single web pages recorded by the website in October 2019, November 2019 and January 2020 could be due to the flu season, with the related search for information on the flu vaccination campaign. Another possible reason could be the need to find information on immunization certificates by parents for the admission of their children to school and the catch-up of the missing vaccinations, according to the mandatory national law [20].

The majority of website visitors and Facebook followers are young females. This result could be related to the growing trend of women searching for information about immunization and vaccination on the internet. Indeed, in most families, women are still the children’s principal care-givers as regards health issues such as vaccination in childhood. Moreover, the Italian Ministry of Health endorsed two documents (in 2018 and in 2019) [21,22] including recommendations on vaccinations for pregnant women and/or in childbearing age. Actually, pregnancy is among the most critical moments in a woman’s life, during which worries, and responsibilities related to the health of life in her womb increase, as do fears of taking drugs with potential undesirable effects on the fetus. On the contrary, the administration of vaccines during pregnancy is linked to the protection of the woman and her child [21]. Randomized controlled trials showed that providing information on immunization through websites with interactive social media components during pregnancy can positively influence parental vaccine decisions [23]. A pilot study on the impact assessment of an education course on vaccinations in a population of pregnant women showed that the number of women who rated their level of knowledge of vaccinations as poor or insufficient had decreased by 30% after the intervention, while the number of “hesitant” respondents decreased, especially when the decision to be vaccinated during pregnancy was analyzed [24].

Our results suggest that the Facebook account launch could have contributed to the increase in the visibility of the website. The use of social networking sites continues to grow worldwide to share contents and to learn. Social networks are used by millions of people every day to interact and engage with other users. Social networking sites provide an immediate and personal way to deliver program, products and information. The most popular social networking site is Facebook, which has over 750 million users. The average user creates 90 pieces of content every month, and 50% of active users log on to the site on any given day. There are several public health-related Facebook sites available with different targets, purposes and functions [25]. Facebook is a public platform and, in most cases, reaches the general public. Specifically, targeted Facebook pages can be developed to address healthcare providers, public health professionals and others. The US Centers for Disease Control and Prevention (CDC) launched the Facebook page “Parents are the Key to Safe Teen Drivers” as example of new communication approach targeted to parents of teenagers. There are many Facebook pages dedicated to public health topics; in particular, with regard to vaccinations, many pages deal with this topic (“Vaccine info”, “Vaccines save lives”, “Rete Informazione Vaccini”, “Iovaccino”).

As an example, online advertising and social media were used to promote a dissemination project on a vaccine-preventable disease in Italy, the PneumoRischio eHealth project. The app PneumoRischio, with a risk checker that allowed the calculation of the risk of contracting pneumococcal diseases, and a related website and a Facebook account were created in order to increase the knowledge of invasive pneumococcal disease and the awareness of the existence of a vaccine that prevents it [26]. The marketing campaign, started 7 months after release, significantly increased the views [27].

Marar et al. prove that the majority of the participants used social media platforms to find information related to their health conditions, while approximately one third received direct medical consultations online. However, public awareness to use reliable sources for health information is needed [28]. In this regard, healthcare workers, such as General Practitioners and pediatricians, could represent the key role in modifying personal opinions and choices about vaccinations. A proper vaccination counseling could enhance the way people approach vaccination issues, such as the authority of web-based contents [29]. Social networking services are major channels of health communication in responding to issues on infectious diseases and other health problems. New videos posted on the internet attract considerable amounts of attention from social network users and increase traffic to certain web sites [30]. As the public increasingly uses the tools of the internet and social media, scientists and healthcare workers also need to use old and new ways of communication to fight the phenomenon of hesitation with information of high quality [31].

There is some evidence that text messaging, accessing immunization campaign websites, using patient-held web-based portals and computerized reminders increase immunization coverage rates. Insufficient evidence is available on the use of social networks, email communication and smartphone applications [32].

In our study, we showed an increase in the visibility of the VaccinarSinToscana website and the related Facebook account, but we cannot demonstrate a link with the increase in the immunization coverage rates in Tuscany. After the Italian mandatory law on immunization for school attendance, the immunization coverage increased in our region. Probably the raise in the number of visits to our website could also be related to the endorsement of the national law.

The objective for the future of the VaccinarSinToscana portal is to grow further to reach out to an even wider audience. Moreover, the portal could take advantage of further collaborations with national and local agencies and scientific associations. Finally, innovative communication media and skills (i.e., Twitter and Instagram account) could be useful to reach the overall aim of the VaccinarSì Project to further promote the culture of immunization and counteract the alarming phenomenon of vaccine hesitancy.

## 5. Conclusions

The experience with the VaccinarSinToscana portal, together with the other national and regional channels, shows that institutional websites and social networking with evidence-based information could be useful tools for users and health professionals in order to reach the empowerment of the population in making aware choices on immunization and the objectives of public health.

## Figures and Tables

**Figure 1 vaccines-08-00644-f001:**
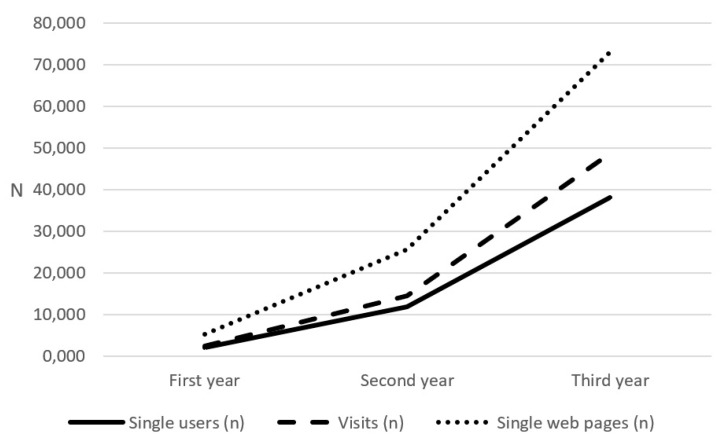
Number of annual singles users, visits and single web pages visited during the three years of the study (first year: 21st March 2017–20th March 2018; second year: 21st March 2018–20th March 2019; third year: 21st March 2019–20th March 2020).

**Figure 2 vaccines-08-00644-f002:**
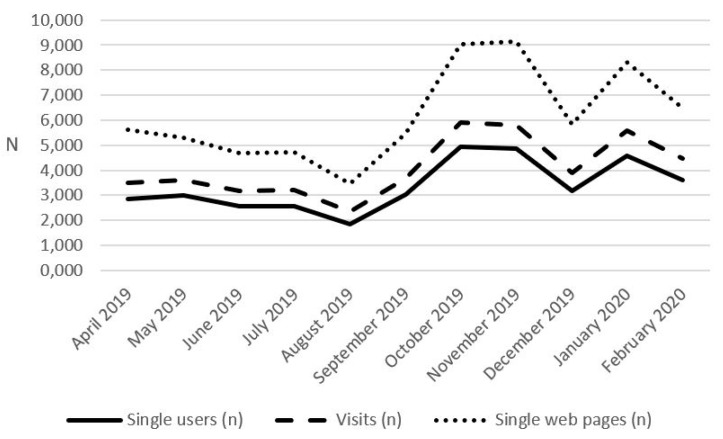
Number of monthly single users, visits and single web pages from April 2019 to February 2020.

**Figure 3 vaccines-08-00644-f003:**
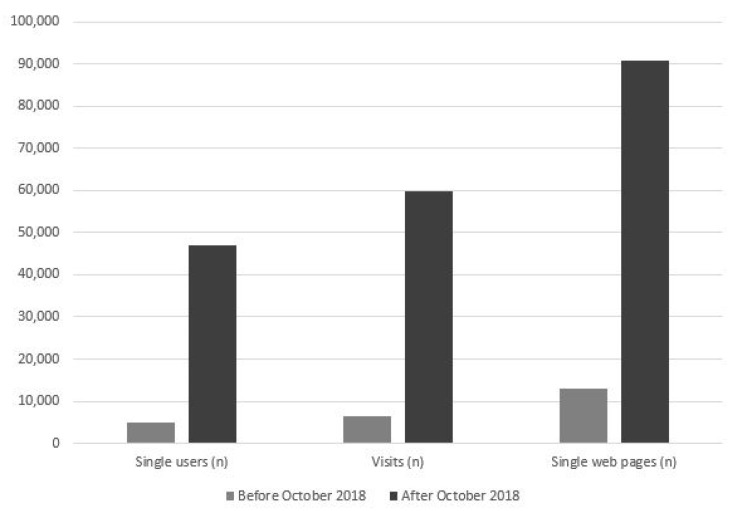
Number of annual singles users, visits and single web pages before and after Facebook account was launched in October 2018 (before October 2018 = 558 days; after October 2018 = 536 days).

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
