# Peer review of "The Experience of VaccinarSinToscana Website and the Role of New Media in Promoting Vaccination"

_vaccines, 2020, doi:10.3390/vaccines8040644_

Round 1

Reviewer 1 Report

The study attempts to analyze three years of activity on the VaccinarSinToscana website and its related Facebook page. Communicating about vaccination via websites is increasingly important these days. Although I find the topic very interesting, the study seems to function as a report rather than an academic study. I also have the following specific concerns.

The introduction provides some necessary information regarding the development of the VaccinarSinToscana website. However, the authors need to highlight the importance of this research and the contributions they expect the study to make.

The authors need to further develop the study’s conceptual framework. For example, the authors state that their aim is “to present the VaccinarSinToscana website framework and analyze the three-year activity of the website and the related social network account on Facebook in terms of dissemination and visibility”; however, what they mean by “dissemination and visibility” remains unclear.

The introduction also does not summarize relevant prior research.

For readers to understand the study, the methods section needs to include more details. The section only briefly explains how the data were collected, and replicating the process would be difficult based on this description. Much more elaboration is needed.

The results section is interesting, but it does not seem to provide answers to the study’s research question. Indeed, the results and discussion seem to be irrelevant to the research question. This may be partly a result of the unclear nature of the research question.

I hope the authors find my comments helpful in improving their manuscript. Thank you for giving me the opportunity to review this study.

Author Response

Reviewer 1: The study attempts to analyze three years of activity on the VaccinarSinToscana website and its related Facebook page. Communicating about vaccination via websites is increasingly important these days. Although I find the topic very interesting, the study seems to function as a report rather than an academic study. I also have the following specific concerns:

Point 1: The introduction provides some necessary information regarding the development of the VaccinarSinToscana website. However, the authors need to highlight the importance of this research and the contributions they expect the study to make.

Response 1: We modified the aim of the study in order to better specify the importance of this research in the dissemination of scientific information about vaccinations, both nationally and especially locally. We also added the contributions we expect from the study. We expect that more and more users will access the website VaccinarSinToscana in the future and that the website will become an example of an active tool to spread correct information about vaccinations in the general population (lines81-88).

Point 2: The authors need to further develop the study’s conceptual framework. For example, the authors state that their aim is “to present the VaccinarSinToscana website framework and analyze the three-year activity of the website and the related social network account on Facebook in terms of dissemination and visibility”; however, what they mean by “dissemination and visibility” remains unclear.

Response 2: Dissemination and visibility were evaluated by describing and analysing the number and the socio-demographic characteristics of single users who accessed the website, the number of visits and single web pages visited, and the number of followers of the related Facebook account, during the three-year period of the study (lines83-85).

Point 3:The introduction also does not summarize relevant prior research.

Response 3:We reported the experiences of two international and national projects: Vaccine Today and Esculapio project. Both projects highlight the importance of using new mass media, particularly Internet and social networks, in the communication and dissemination of scientific information both for the general population and for the healthcare workers. (lines 52-58).

Point 4: For readers to understand the study, the methods section needs to include more details. The section only briefly explains how the data were collected, and replicating the process would be difficult based on this description. Much more elaboration is needed.

Response 4: We updated the Materials and Methods section with more detailed information (lines92-97).

Point 5: The results section is interesting, but it does not seem to provide answers to the study’s research question. Indeed, the results and discussion seem to be irrelevant to the research question. This may be partly a result of the unclear nature of the research question.

Response 5: We specified the research question in the introduction, as detailed in the previous points. We specified our results in terms of single users, visits and single web pages visited (line 225-226).

Reviewer 2 Report

I have read this paper with great interest, and consider this paper as 'an example' on how information can be provided, and how its use can be assessed. 

I do have two lines of comments

a fist one, how does this compare to other papers in the literature on website use for vaccine information, or related topic. At present, this paper still reads very much as a single, stand alone document. 

secondly, some specific suggestions 

Introduction, line 37 onwards

I agree on the concept of vaccine hesitance, but the absolute risk for an adverse reactions versus the absolute risk of an adverse event related to the infection does alter for the majority of vaccines currently introduced once a high vaccination coverage has been attained, so that the absolute individual benefit will be lower.

As non-Italian, can the authors further explain the benefit of regional websites ? does this relate to prevention responsibilities at the level of the regions, or are the differences in vaccination scheduled in between regions ? do these different website contain the same information ? If so, how does the Toscan ‘social media’ activities compare to other regional websites, and is there a structure to share expertise and experiences between the different regional websites.

Is there a conflict of interest procedure for the collaborators, and if so, is this open to the public ?

What’s the targeted readership, the public, or also healthcare providers ?

I assume that the users (line 124) differ from the ‘general characteristics’ of the population, but how well do they reach the targeted readership ?

The effect of the facebook account launch is associated with more traffic, but this is not necessary a causal relationship (although reasonable assumption, but methodological this is an association)

Author Response

I have read this paper with great interest, and consider this paper as 'an example' on how information can be provided, and how its use can be assessed. 

I do have two lines of comments:

Point 1: a fist one, how does this compare to other papers in the literature on website use for vaccine information, or related topic. At present, this paper still reads very much as a single, stand alone document. 

Response 1: We added the experience of the ESCULAPIO project, funded by the Italian Ministry of Health (lines 55-58).

Point 2: I agree on the concept of vaccine hesitance, but the absolute risk for an adverse reactions versus the absolute risk of an adverse event related to the infection does alter for the majority of vaccines currently introduced once a high vaccination coverage has been attained, so that the absolute individual benefit will be lower.

Response 2: This is a personal view of the reviewer. In the manuscript, we reported 4 papers clarifying this concept (references 5-8).

Point 3: As non-Italian, can the authors further explain the benefit of regional websites? does this relate to prevention responsibilities at the level of the regions, or are the differences in vaccination scheduled in between regions? Do these different website contain the same information? If so, how does the Toscan ‘social media’ activities compare to other regional websites, and is there a structure to share expertise and experiences between the different regional websites.

Response 3: All regional websites share some information with the national VaccinarSi website (i.e. infectious diseases fact sheets and summary of the vaccine characteristics). Each regional website reports regional newsand events, regional vaccination prevention policies, information about local vaccination clinics (sites, contacts and schedule time for access). Since 2001, Italian Regions have autonomy in health system management (lines 69-72), thus each region endorses a regional plan for immunization. The scientific experts of the regional websites coordinate with the responsible of the national website, and the coordination committee carries out some monitoring on regional websites to increase the activity. There is no available information already published by the other regional websites that allows a comparison with our study, but a specific paper is in progress.

Point 4:Is there a conflict of interest procedure for the collaborators, and if so, is this open to the public?

Response 4: The experts collaborate on a voluntary basis and there is no conflict of interest in this activity (lines 103-104). The website is open and free accessible to the public (line 118).

Point 5:What’s the targeted readership, the public, or also healthcare providers?

Response 5: The website is open and free accessible to the public. Thus, there is no targeted readership. The website addresses to the entire population. The information can be useful also for the healthcare providers (as specified in lines 74-75).

Point 6: I assume that the users (line 124) differ from the ‘general characteristics’ of the population, but how well do they reach the targeted readership?

Response 6: As already specified, there is no targeted readership. We did not aim to evaluate a relationship between the characteristics of the general population and the users of the website. Moreover, no information is collected on the users' profession and so we cannot distinguish between health professionals and the general population.

Point 7: The effect of the Facebook account launch is associated with more traffic, but this is not necessary a causal relationship (although reasonable assumption, but methodological this is an association).

Response 7: We agree with the reviewer. We can only suggest that the Facebook account launch could have contributed to increase the visibility of the website (line 256).

Reviewer 3 Report

This interesting study has introduced the experience of using a website to promote vaccination. In an era when social media is booming, leveraging these new channels to promote vaccination seems promising. I have two comments which might be of some help.

1. Figure 1 has presented the increase of visits in the three years. Would the authors please present the results by month so that readers could see if there was any seasonal change in the number of visitors? Some diseases have seasonal pattern, so in some seasons people might be more interested in searching for vaccine information;

2. Since the authors have the visits’ socio-demographic information, it may be worth it to stratify the analyses in Figures 1 and 2 by socio-demographic characteristics. By doing so, the authors would most likely have an in-depth understanding of which group(s) had a sharper increase than the other groups.

Author Response

Point 1: Figure 1 has presented the increase of visits in the three years. Would the authors please present the results by month so that readers could see if there was any seasonal change  in the number of visitors? Some diseases have seasonal pattern, so in some seasons people might be more interested in searching for vaccine information

Response 1: We thank the reviewer and we added a graphic in order to show the monthly trend of number of single users, visits and single web pages during the study period (lines 175-182 and 236-240). 

Point 2: Since the authors have the visits’ socio-demographic information, it may be worth it to stratify the analyses in Figures 1 and 2 by socio-demographic characteristics. By doing so, the authors would most likely have an in-depth understanding of which group(s) had a sharper increase than the other groups.

Response 2: Google analytics does not allow a stratified analysis by gender and age group.